# Learning Dynamics Feature Representation via Policy Attention for Dynamic Path Planning in Urban Road Networks

**Kai Zhang**[1], **Jingjing Gu**[1,2,*], **Qiuhong Wang**[1]
[1]Nanjing University of Aeronautics and Astronautics, Nanjing, China
[2]National Key Laboratory of Helicopter Aeromechanics, Nanjing, China
[*]Corresponding author
{zhangkainone,gujingjing,wangqiuhong}@nuaa.edu.cn

## Abstract

Dynamic Path Planning (DPP) in urban road networks is challenging due to rapidly changing traffic conditions that often invalidate pre-planned routes. Reinforcement Learning (RL) can adaptively handle such dynamics, but its effectiveness depends on how those dynamics are represented in the state. Existing methods either rely on global dynamics, which are complete but computationally expensive, or on simplified local dynamics, which are efficient but may omit critical information, leading to suboptimal routes. To address this trade-off, we propose a Dynamics Feature Representation (DFR) framework that progressively refines high-dimensional global dynamics into compact, decision-relevant features. DFR introduces policy attention mechanism to identify a core subset of dynamics based on distance-oriented policy, and uses $n$-hop neighborhoods method to further decouples this subset into node-related local feature sequences. This hierarchical refinement enables efficient near-optimal policy learning. Experiments on realistic urban graphs show that DFR improves the performance of RL-based DPP models and accelerates convergence compared to baselines. This work highlights the importance of adaptive state representation and provides a general framework for dynamic decision-making in practical urban transportation.

## 1 Introduction

With the rapid development of emerging applications such as on-demand delivery, urban logistics, and intelligent transportation systems, achieving efficient path planning in urban road networks has become a pressing demand (Gu et al., 2023; Mao et al., 2025). However, real-world traffic conditions are time-varying, often rendering planned routes ineffective. Dynamic Path Planning (DPP) has thus been proposed to address this issue (Du et al., 2024a), aiming to maintain robust and adaptive routing under continuously evolving traffic conditions. A common way to formulate DPP is to model the urban road network as a dynamic directed and weighted graph. Traditional methods then rely on explicit traffic flow prediction or rule-based evolution models (Guo et al., 2024) (e.g., forecasting link speeds for a future time horizon). However, such predictions suffer from limited accuracy and poor generalization, particularly under unexpected events like accidents or road closures (Medina-Salgado et al., 2022; Huang et al., 2022). Therefore, those approaches may be unreliable in practice.

Reinforcement Learning (RL) offers a new paradigm for handling dynamics. Instead of requiring an accurate explicit model, RL incorporates dynamics into the state and learns optimal decisions through interaction (Bouktif et al., 2023; Lin et al., 2025). The key advantage is that value function estimation implicitly absorbs the statistical patterns of state transitions, thereby reducing reliance on predictive accuracy and exhibiting stronger robustness to unseen events (Nguyen et al., 2025). Despite this advantage, applying RL to DPP faces a fundamental challenge (Chen et al., 2023): *how to effectively represent traffic dynamics effectively within the state*. Existing approaches struggle with a dilemma. Global methods that encode the entire graph dynamics to ensure information completeness but are computationally prohibitive Liu et al. (2024). Local methods that use partial observations of the agent are efficient but risk overlooking essential non-local dynamics (Du et al.,

2024b), leading to shortsighted and suboptimal routing (Francis et al., 2025). More critically, insufficient state representation may undermine the Markov property, which leads to degraded model performance and unstable training. Therefore, it is crucial for effective RL-based DPP to identify suitable dynamics features.

To bridge this gap, we propose a Dynamics Feature Representation (DFR) framework to construct a Markovian state representation by progressively distilling the global graph dynamics into a compact, decision-centric feature. DFR employs a novel hierarchical refinement process: 1) **Task-related filtering via policy attention:** We first pre-train a policy attention module that is expert in finding shortest paths. Given an source and destination, it identifies the top-$k$ shortest paths and extracts a sparse, task-relevant subgraph from the global graph based on the nodes in those paths. The dynamics of this subgraph form the first refined state. 2) **Agent-centric decoupling via $n$-hop neighborhoods:** The above state is further refined at each timestep to focus on the agent's immediate context. We compute the $n$-hop neighborhoods around the current node and take its intersection with the policy attention subgraph. The resulting dynamics provide a low-dimensional state vector that captures the essential local traffic conditions within the context of the global task. By doing so, DFR provides the RL agent with a state that is both computationally efficient and informationally sufficient to approximate the Markov property.

Our core contributions are summarized as follows: 1) We propose the DFR framework, a novel hierarchical approach to resolve the completeness-efficiency trade-off in state representation for RL-based DPP. 2) We introduce two key technical innovations within DFR: a policy attention mechanism for task-aware graph sparsification that extracts a task-related subgraph, and a $n$-hop neighborhood method for agent-centric feature decoupling that provides a local state view. 3) We empirically validate that our DFR framework achieves a significant improvement in performance and a remarkable acceleration in convergence compared to standard baselines.

## 2 RELATED WORK

**Path planning methods** Traditional path planning algorithms, such as A* (Hart et al., 1968) and D* Lite (Koenig & Likhachev, 2002), have been widely used in static environments. However, their reliance on a pre-defined and fixed cost map makes them ill-suited for highly dynamic urban networks where travel costs change rapidly. To address this, some learning-enhanced methods have been proposed (Du et al., 2024a; Li et al., 2024). These typically involve first predicting future traffic conditions using statistical or deep learning models (Guo et al., 2024; Yang et al., 2025), and then executing a classic search algorithm on the predicted graph. However, the performance of these planners is inherently bounded by the accuracy of the traffic forecast (Tian et al., 2025). They are particularly vulnerable to distribution shift and unexpected events (e.g., accidents), often leading to suboptimal or failed routes (Ren et al., 2025). In contrast, our method eliminates the need for explicit prediction, and we employ RL to learn a policy that implicitly captures the statistics of traffic dynamics through interaction, thereby achieving greater adaptability.

**State representation** RL has emerged as a powerful framework for solving DPP problems (Mao et al., 2023; Xue et al., 2025). A common challenge is the design of the state representation (Francis et al., 2025). In the context of path planning on graphs, previous work often resorts to simplistic representations. Some methods use a local view (Zhao et al., 2025), which sacrifices global optimality for efficiency. Others provide a global view of the entire network (Lin et al., 2025), which is computationally prohibitive in large cities. In addition, graph neural networks offer a powerful way to process graph-structured data for DPP problem (Zang et al., 2023; Sun et al., 2025), by encoding the graph structure and dynamics into node embeddings. While they can theoretically capture the entire graph's information, the computational and memory overhead scale with the size of the graph (Wu et al., 2020), making them impractical for real-time planning in massive urban networks. The state for decision-making is often assumed to be given and Markovian, which is frequently not the case in practice (Kaelbling et al., 1998). Our work attemps to refine global dynamics into a compact form, drawing inspiration from feature decorrelation techniques (Huang et al., 2023).

**Attention mechanisms** The use of attention mechanisms for state abstraction and feature selection is well-established (Brauwers & Frasincar, 2021; Hu et al., 2024). Soft attention, as in Transformers (Min et al., 2022), allows agents to weight the importance of different parts of their input. Our

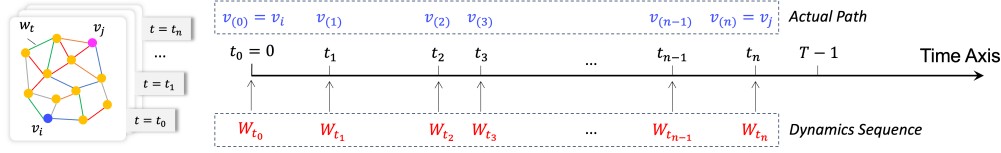

Figure 1: The discrete timeline of dynamics in urban road network system: ranging from 0 to $T-1$, each time $t_k \in \{0, 1, \ldots, T-1\}$ corresponds to a node $v_{(k)}$ in the path, where $k = 0, 1, \ldots, n$.

proposed policy attention is a hard, pre-computed attention based on the structural semantics of the task. This provides a strong, interpretable prior that drastically reduces the problem's dimensionality before the RL agent even begins learning, similar in spirit to knowledge distillation approaches used in other domains (Papadopoulos et al., 2021).

## 3 PROBLEM FORMULATION

Next, we formally describe the DPP problem using graph theory and Markov decision process.

### 3.1 DYNAMIC PATH PLANNING IN DISCRETE URBAN MAPS

Assuming that the road network of a city is represented as a directed and weighted graph $G = (V, E)$, where $V = \{v_1, v_2, \ldots, v_N\}$ is the node set, and $E = \{e_{i,j} | v_i, v_j \in V, i \neq j\}$ is the edge set. Each $e_{i,j} = (v_i, v_j)$ has a connection value $l(v_i, v_j)$ and a weight value $w(v_i, v_j; t)$, where $l(v_i, v_j) = \{0, 1\}$ represents the connection relationship, and $l = 1$ indicates that node $v_i$ and $v_j$ are directly connected and $l = 0$ means no direct connection; and $w(v_i, v_j; t) \in \mathbb{R}$ denotes the traffic cost (such as time or distance) from $v_i$ to $v_j$ at time $t$. For all $t$, $|w(v_i, v_j; t)| < \infty$ if $l(v_i, v_j) = 1$, and $|w(v_i, v_j; t)| = \infty$ if $l(v_i, v_j) = 0$.

It is assumed that $V$ remains constant, while the weight $w(\cdot, \cdot; t)$ of each edges may vary over time $t$. The set of edge weights $W_t : \mathbb{R}^{|E|} = \{w(v_i, v_j; t) \mid \forall (v_i, v_j) \in E\}$ is referred to as the *dynamics* of the urban road network system at time $t$. Furthermore, we define the discrete sequence of dynamics over a time horizon $T$ as $\mathbf{W}_{:T} = \langle W_0, W_1, W_2, \ldots, W_{T-1} \rangle$, which characterizes the temporal evolution of the system. We define the path space $\mathcal{P} = \{p\}$, and path $p$ can be described as a sequence of nodes, and adjacent nodes are directly connected, that is, $p = \langle v_{(0)}, v_{(1)}, \ldots, v_{(n)} \rangle_{1 \leq n < \infty}$, where the $k^{\text{th}}$ node in $p$ can be represented as $p_k = v_{(k)}$ and $l(p_k, p_{k+1}) = 1, k = 0, 1, \ldots, n-1$. Given a source node $v_i \in V$ and an goal node $v_j \in V$, the path space between them is defined as $\mathcal{P}[v_i, v_j] \subset \mathcal{P} = \{p[v_i, v_j]\}$, and $p_0 = v_i, p_n = v_j$ for each path $p[v_i, v_j]$. Figure 1 shows the correspondence between nodes in the path and dynamics sequence on the timeline.

The DPP problem aims to find the optimal path with minimum traffic cost between any node pair under changing traffic dynamics [1], formulated as a combination optimization problem over discrete path spaces. Given a source $v_i$ and a goal $v_j$, along with the temporal dynamics $\mathbf{W}_{:T}$, the DPP task is defined as $\tau = (v_i, v_j; \mathbf{W}_{:T})$. Aassuming that $\mathbf{W}_{:T}$ is known in advance, the traffic cost of a path $p$ under $\mathbf{W}_{:T}$ is denoted as $c(p; \mathbf{W}_{:T})$. The optimal path between nodes $v_i$ and $v_j$ can be written as

$$p^*[v_i, v_j; \mathbf{W}_{:T}] = \arg\min_{p \in \mathcal{P}(v_i, v_j)} c(p; \mathbf{W}_{:T}) = \arg\min_{p \in \mathcal{P}(v_i, v_j)} \sum_{k=1} w(p_k, p_{k+1}; t_k) \tag{1}$$

This is an ideal strategy that leverages the evolution of dynamics, enabling the planner to reason about the long-term consequences of current paths and thus achieve global optimality. However, in practice, future dynamics are typically unknown or inaccurate, making it a theoretical benchmark for evaluating other strategies. To address the limitations, Markov Decision Process (MDP) is introduced to rigorously reason about dynamics patterns by leveraging probability theory and stochastic process and learn near-optimal policies without requiring knowledge of future traffic conditions.

---

[1]It is worth noting that "dynamic" refers to the replanning of paths, while "dynamics" refers to the changes of traffic conditions in the urban road network. Dynamics is the direct cause of dynamic path planning.

## 3.2 MARKOV DECISION PROCESS

**Agent and environment**  Assuming that the vehicle is abstracted as a RL agent, and the DPP problem in urban graph can be mapped into the finite-horizon and deterministic MDP model as a 6-tuple $\mathcal{M} = (\mathcal{S}, \mathcal{A}, T, R, \gamma, H)$:

State space $\mathcal{S}$ denotes the set of all possible states, and a state $s_t \in \mathcal{S} = \{v^t, v_g, f_t\}$ consists of the current node $v^t$ (with $v^0 = v_i$ as the source node) [2], the goal node $v_g = v_j$, and the *dynamics features* $f_t$, where $(v_i, v_j; \mathbf{W}_{:T})$ is an instance of the DPP task. $f_t$ encodes current traffic conditions observable by the agent, which is a *compressed representation* of $\mathbf{W}_{:T}$ that provides environment perception relevant to the current decision. Action space $\mathcal{A} \subseteq \mathbb{R}^m$ is the set of possible actions taken by the agent, and the action $a_t = \{0, 1, \ldots, n_a\}$ is designed to move to one of $n_a$ neighbors of $v_a^t$ in the graph. Transition function $T : \mathcal{S} \times \mathcal{A} \times \mathcal{S} \to [0, 1]$ defines the probability of transitioning from current state $s_t$ to next state $s_{t+1}$ after taking action $a_t$, that is, $T(s_{t+1}|s_t, a_t)$.

Reward function $R : \mathcal{S} \times \mathcal{A} \times \mathcal{S} \to \mathbb{R}$ guides the agent to achieve the goal by assigning a numerical reward to its each transition $(s_t, a_t, s_{t+1})$. In this paper, $R$ is the negative value of the traffic cost under dynamics $W_t$ between current node $v^t$ in $s_t$ and next node $v^{t+1}$ in $s_{t+1}$,

$$R(s_t, a_t, s_{t+1}) = -c(v^t, v^{t+1}; W_t) + b \cdot \mathbb{I}(v^{t+1} == v_g) \tag{2}$$

where $b \in \mathbb{R}^+$ is a constant reward for reaching the goal node, and $c(\cdot, \cdot)$ is the traffic cost of two nodes (see Equation 1). Discounted factor $\gamma \in [0, 1]$ is used to balance between the future reward ($\gamma \to 1$) and immediate reward ($\gamma \to 0$). Time horizon $H \in \mathbb{N}^+$ is the maximum number of steps, that is, $t \leq H$. If the agent cannot reach the goal within $H$ steps, the task ends in failure.

**Policy and value function**  Policy function $\pi : \mathcal{S} \to \Delta(\mathcal{A})$ defines a strategy of the agent to decide the next action based on the current state. Given a DPP task $\tau$ with the initial state $s_0$, a path or state sequence $p = \{(s_t, a_t, s_{t+1})_{t=0}^{h-1}\}_{h \leq H}$ can be automatically generated by $a_t \sim \pi(\cdot|s_t)$ and $s_{t+1} \sim T(\cdot|s_t, a_t)$. The ultimate goal is to obtain the optimal policy $\pi^*$ that can derive the optimal path $p^*$. To quantitatively evaluate the utility of $\pi$, the state value function $V : \mathcal{S} \to \mathbb{R}$ and the state-action value function $Q : \mathcal{S} \times \mathcal{A} \to \mathbb{R}$ are introduced, which is the expectation of accumulative rewards under $R$ at state $s$ that the agent can obtain by following $\pi$ and $T$. For the convenience of calculation, $V$ can be transformed into a recursive form, which is the famous Bellman Equation,

$$V_\pi(s) = \sum_{a \in \mathcal{A}} \pi(a|s) Q_\pi(s, a) = \sum_{a \in \mathcal{A}} \pi(a|s) \sum_{s' \in \mathcal{S}} T(s'|s, a) \left[ R(s, a, s') + \gamma V_\pi(s') \right] \tag{3}$$

**Solution optimality**  The decision condition that a policy is better than another policy is converted to the ordering relationship between their value functions. The optimal policy $\pi^*$ is the policy corresponding to the upper bound of $V$, e.g. $\pi^* = \arg\max_{\pi \in \Pi} V_\pi(s), \forall s \in \mathcal{S}$, and the Bellman Optimality Equation is obtained, that is,

$$V_{\pi^*}(s) = V^*(s) = \max_a \sum_{s'} T(s' \mid s, a) \left[ R(s, a, s') + \gamma V^*(s') \right] \tag{4}$$

Using classic DRL algorithms, such as Deep Q-Network (DQN) (Mnih et al., 2015) and its variants (Van Hasselt et al., 2016; Wang et al., 2016), and Proximal Policy Optimization (PPO) (Schulman et al., 2017), the optimal value function $V^*(s)$ can be uniquely determined by Equation 4. Even in large-scale state spaces, DQN leverages neural networks to parameterize the value function and employs sample-based updates to gradually improve its estimates until convergence to $V^*(s)$.

## 4 METHODOLOGY

### 4.1 CHALLENGES

We first analyze the key challenges in constructing effective state representations in the MDP-based DPP model. In Section 3.2, the state is $s_t = \{v^t, v_g; f_t\}$, where $f_t$ encodes the dynamics features

---

[2]The $t$ here is different from the $t$ in the previous part of graph theory. Simply put, $t = n$ here represents the $n^{\text{th}}$ decision step, which corresponds to $t = t_n$ on the time axis in graph theory part.

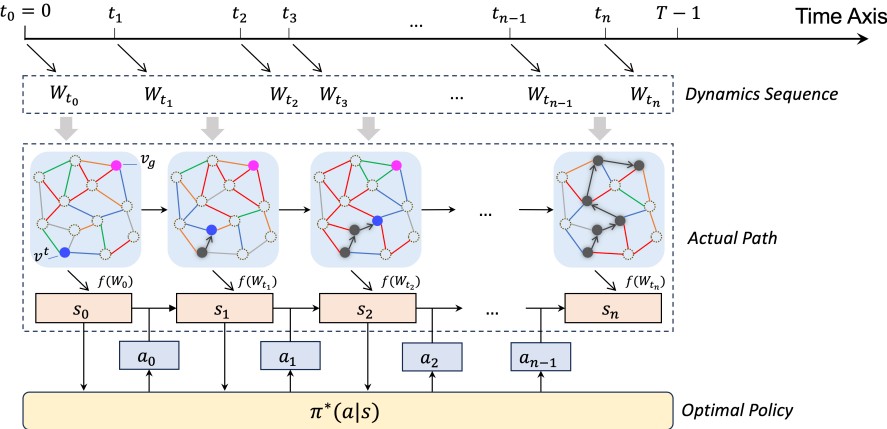

Figure 2: The diagram illustrating how the policy function handle dynamics sequence in MDP-based DPP. The colors of the edges in the graph represent different dynamics.

extracted from the environment (see Figure 2). The quality of $f_t$ directly determines the agent's planning ability to reason about dynamic traffic conditions. However, the design of $f_t$ faces an inherent trade-off. When $f_t$ is global dynamics, the formulation resembles a fully observable MDP, where the agent has access to the complete system state. When $f_t$ is limited to local dynamics, the formulation becomes partially observable, in which the agent act under incomplete and uncertain knowledge of the environment. Such partial observability reflects real-world conditions, where future states are inherently unavailable. Therefore, the major challenge of dynamics feature design is *sufficient yet compact*: retaining the essential dynamics for decision-making while avoiding redundancy and excessive complexity. Therefore, we propose a *Dynamics Feature Representation (DFR)* framework, which aims to identify and encode task-relevant subsets of dynamics, striking a balance between information sufficiency and computational tractability.

## 4.2 DYNAMICS FEATURE REPRESENTATION

Given a DPP task $\tau = (v_s, v_g; \mathbf{W}_{:T})$, and the state representation $s_t = \{v^t, v_g; f_t\}$. We formalize DFR as a three-level process, refining the global information into a compact, decision-relevant state.

$$\mathbf{W}_{:T} \xmapsto{\tau, \Psi} \mathbf{W}'_{:T} \xmapsto{v^t, \Phi} \mathbf{W}''_{:T} \qquad (5)$$

**Global dynamics features W** At decision step $t$, the full graph's edge weights $W_t$ serve as the dynamics feature $f_t$, that is, $s_t = (v^t, v_g; W_t)$, where $v^t$ is the current node. As mentioned before, although $W_t$ is extremely high-dimensional and redundant, making it unsuitable to be used directly as the state input for a RL agent. $W_t$ then provides the raw input for subsequent hierarchical processing to produce more compact, task-relevant, and node-specific features.

**Task-related key features W′** To reduce redundancy while retaining essential information, we define a global key subset $W'_t(\tau) = \Psi(\tau, W_t)$, where $\Psi$ is a function that extracts partial dynamics from $W_t$, that is, $W'_t \subseteq W_t$. $W'_t$ significantly reduces the dimensionality while preserving the critical dynamics information required for optimal routing decisions. Formally, for task $\tau$, $W'_t$ is sufficient if the optimal policy conditioned on it approximates the policy conditioned on $W_t$:

$$\pi^*(v^t, v_g; W'_t) \approx \pi^*(v^t, v_g; W_t) \qquad (6)$$

**Node-related local features W″** Even after global filtering, $W'_t$ may remain high dimensionality in large-scale DPP scenarios. Therefore, we further define node-related local features $W''_t(v^t) = \Phi(W'_t, v^t)$, where $\Phi$ is a feature extraction function that maps $W'_t$ to a lower-dimensional subset $W''_t$ associated with the current node $v^t$. $W''_t$ can incorporate a spatial and temporal neighborhood, forming a time-aware feature sequence. Similarly, the optimality of policy is preserved, that is,

$$\pi^*(v^t, v_g; W''_t) \approx \pi^*(v^t, v_g; W'_t) \qquad (7)$$

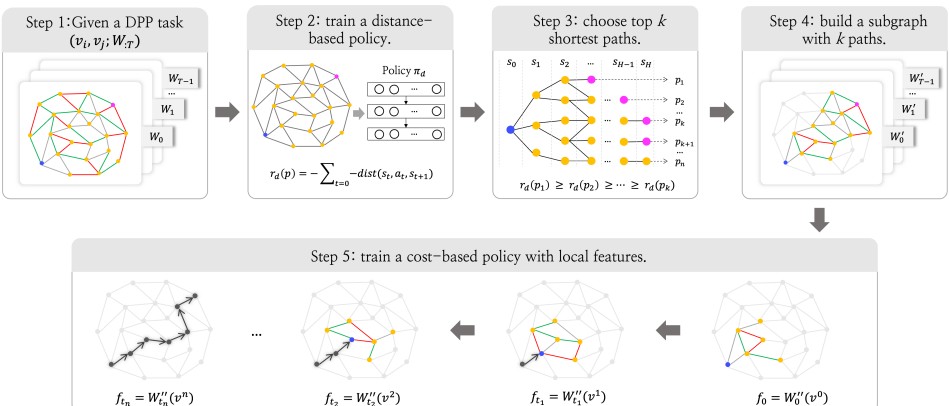

Figure 3: The diagram of dynamics feature design using policy attention mechanism and $n$-hop neighborhood method ($n = 2$ is shown in the example). The colors of the edges in the graph represent different dynamics.

**Theoretical Basis.** The local feature $f_t = W_t''$ must ensure the *compactness* of the state representation, meaning that it preserves all decision-relevant information from the original dynamics. Predictive State Representations (PSR) provide a theoretical foundation for this requirement (Littman & Sutton, 2001). PSR posits that the state of a system can be defined by predictions of future observable outcomes given possible action sequences, without resorting to latent variables. In this sense, $W_t''$ serves as a predictive representation of the state: it encodes sufficient information to forecast the effects of future actions, thereby ensuring that the optimal policy based on $W_t''$ approximates the policy based on the full dynamics sequence $\mathbf{W}_{:T}$:

$$\pi^*(v^t, v_g; W_t'') \approx \pi^*(v^t, v_g; \mathbf{W}_{:T}) \tag{8}$$

Beyond spatial abstraction, DFR also preserves the *temporal dependencies* inherent in traffic dynamics. Each $W_t$ reflects the evolving traffic state at time $t$, and the refinement process (Equation 5) operates over the sequential structure $\mathbf{W}_{:T}$ rather than on a single snapshot. By filtering and aggregating temporally adjacent representations, DFR implicitly captures short-term temporal correlations—such as local congestion propagation and flow continuity—while maintaining computational efficiency. From the PSR perspective, this design enables $W_t''$ to function as a predictive summary of future dynamics conditioned on the current observation window, thereby preserving both spatial and temporal decision-relevant information. This mechanism aligns with the Markov assumption, which requires the current state to represent all information necessary for decision-making.

Grounding DFR in PSR principles thus guarantees that the resulting representations are compact, temporally predictive, and theoretically sufficient. This not only stabilizes training and mitigates suboptimality under partial observability, but also enhances interpretability and generalization when applying RL to the DPP problem.

### 4.3 DYNAMICS FEATURE EXTRACTION

After formalizing the DFR framework with $\Psi$ for task-related key feature selection and $\Phi$ for node-related local feature extraction, we now instantiate these mappings with concrete methods.

**Policy Attention Method.** Given the global dynamics features $W_t$, the policy attention mechanism is employed to construct a task-relevant key subset $W_t'$ (see Figure 3). It leverages a distance-based optimal policy $\pi_d^*$, which computes the shortest paths from the current node $v^t$ to the destination $v_g$ under static conditions. The paths derived from $\pi_d^*$ are ranked by length, and the top-$k$ shortest paths are selected to form a subgraph $G' = (V', E')$. The edge weights of this subgraph define $W_t'$, where the parameter $k$ controls the trade-off between completeness and compactness: a smaller $k$ may omit critical paths, whereas a larger $k$ may introduce redundant information.

Formally, to obtain $\pi_d^*$, the MDP formulation in Section 3.2 is simplified by removing the dynamics feature component $f_t$ from origin state representation, and the reward function is redefined as

$R_d(s, a, s') = -d(s, s')$, where $d$ denotes the road length between the current state $s$ and the next state $s'$. Substituting the new state representation and reward function $R_d$ into Equation 4, $\pi_d^*$ can be obtained after a certain number of trainning iterations.

As discussed above, DPP aims to find optimal paths under multiple objectives—such as distance, travel time, and energy consumption—by training an RL agent to learn a multi-objective optimal policy. In contrast, the policy $\pi_d^*$ used in the pre-training stage of our DFR framework is distance-based only, without considering temporal variations in edge weights. This corresponds to RL-based static path planning, where the goal is to learn a policy capable of generating shortest paths in a static road network. The reason for using $\pi_d^*$ is that, *even when the ultimate objective of DPP is multi-criteria, distance naturally serves as one of the most fundamental constraints*. Thus, emphasizing edges along the shortest paths ensures that critical dynamics influencing near-optimal planning decisions are captured, making the extraction of $W_t'$ both reasonable and effective. In addition, inter-node distances do not vary over time as long as the topological structure of the urban graph remains constant. Consequently, the pretraining process of $\pi_d^*$ can be one-time and offline, which provides a stable and interpretable reference for subgraph extraction during DPP.

$n$-**hop neighborhood method** Based on $W_t'$, local features are extracted for current node $v^t$ through an $n$-hop neighborhood method. Let $\mathcal{N}^i(v^t)$ denote the $i$-th order neighbors of $v^t$, and $V'$ and $E'$ are the node set and egde set of the policy attention subgraph $G'$, respectively. The local node set of $v^t$ is then defined as $V_l(v^t) = \bigcup_{i=0}^n \mathcal{N}^i(v^t) \cap V'$ and the corresponding edges form the node-related subgraph. The weights of these edges define the local dynamics feature $f_t = W_t''(v^t) = \{w(v_i, v_j; t) \in E' \mid v_i, v_j \in V_l(v^t)\}$. $n$ determines the spatial scale of $W_t''$: smaller $n$ captures highly localized dynamics but may overlook broader context, while larger $n$ expands coverage but increases dimensionality and computational cost.

It is noteworthy that our DFR framework incurs negligible additional computational overhead. The total cost involves updating dynamic traffic information and executing the DPP model. By leveraging policy attention and $n$-hop neighborhoods, DFR reduces the scope of dynamics feature collection to small, pre-computable subgraphs. Both policy attention—derived from a static path planning policy—and $n$-hop neighborhoods depend only on the fixed road network topology, allowing offline computation and reuse. Consequently, DFR achieves significant feature compression without affecting online efficiency, making it suitable for real-world traffic scenarios with live updates.

## 5 EXPERIMENTS

### 5.1 EXPERIMENT SETTINGS

**Scenarios** The experiments are conducted on three urban road networks in China—Nanjing, the Chaoyang District of Beijing, and the Pudong District of Shanghai—sourced from OpenStreetMap (OSM). Each road network for driving is preprocessed and modeled as a directed weighted graph, as illustrated in Figure 4. For each region, we specify a center node in the corresponding network and extract a subgraph by including all nodes within a certain radius. Each scenario corresponds to a single DPP task with a source node, a goal node, and a dynamics sequence. The source and goal nodes are randomly sampled from a subgraph. In this work, we take traffic time as the dynamics, and the path with minimum traffic time is regarded as the optimal path (see Equation 2). Edge weight $w(v_i, v_j; t)$ are parameterized by a congestion factor $\beta(v_i, v_j; t) \in [0.1, 1.5]$. Here, a smaller $\beta$ indicates heavier congestion or higher traffic density, and a larger $\beta$ corresponds to smoother traffic conditions. For instance, $\beta(v_i, v_j; t) = 0.1$ means that the road $(v_i, v_j)$ at $t$ can only be traversed at one-tenth of the base speed. Given the distance $d(v_i, v_j)$ and the base speed $\nu_0$, the traffic time of a road $(v_i, v_j)$ under dynamics $W_t$ is then written by

$$c(v_i, v_j; W_t) = d(v_i, v_j) / (\nu_0 \times \beta(v_i, v_j; t)) \tag{9}$$

**Baselines** We consider three baseline RL algorithms [3]: **DQN** (value-based), **PPO** (policy-gradient, stabilized actor–critic for discrete actions), and **GCN+DQN** (graph-based with GCN feature extraction). Each algorithm is evaluated both with and without DFR. For evaluation, we adopt four metrics:

---

[3]As the advantages of RL-based approaches over traditional methods in DPP have been well established in the literature. Our work instead aim to investigate the impact of the DFR framework within the RL paradigm.

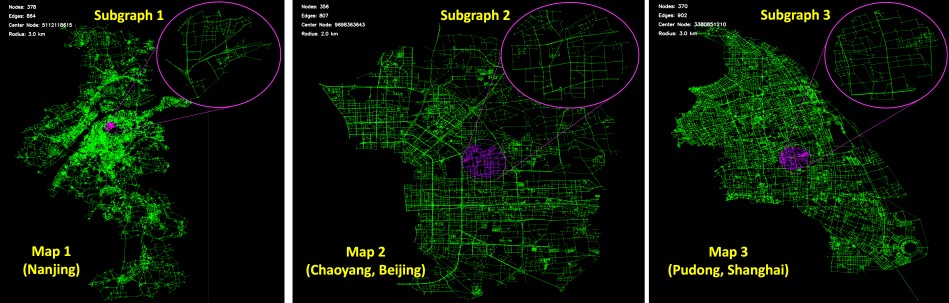

Figure 4: Urban road network of three cities or districts. For each region, a center node and radius are specified, and all nodes and edges within this range are extracted to form a subgraph (magenta).

1) **Mean GAP:** the average relative difference of cost between the learned paths and the ground-truth paths, and lower is better. The ground-truth paths are computed by the dynamic Dijkstra algorithm; 2) **Success Rate (SR):** the proportion of scenarios in which the learned path successfully reaches the goal node, and higher is better; 3) **Compactness Rate (CR):** the proportion of the reduced feature dimension after DFR to the original dimension, and lower is better. **Planning Time (PT):** the average computation time required by an algorithm to generate a complete path for a given planning query, and lower is better.

**Model Training**   Given a subgraph, the DPP model is trained using the above algorithms over $N$ episodes, where each episode corresponds to a new scenario. During training, the DQN updates its Q-function by minimizing the temporal-difference (TD) loss:

$$\mathcal{L}(\theta) = \mathbb{E}_{(s,a,r,s')\sim\mathcal{D}}\left[\left(r(s,a,s') + \gamma \max_{a'} Q(s',a';\theta^-) - Q(s,a;\theta)\right)^2\right],\qquad(10)$$

where $\theta$ and $\theta^-$ denotes the parameters of the Q-network and target network, respectively, $\gamma$ is the discount factor, and $\mathcal{D}$ is the experience replay buffer.

For all experiments [4], we use the Adam optimizer with a learning rate of $10^{-3}$ and a discount factor $\gamma = 0.99$. The number of training episodes is fixed to $75,600$ episodes (about 200 epochs), and the model performance under the above metrics is reported. The size of the replay buffer is $10^6$ and the batch size is 32. Each episode has a fixed horizon of 100 steps. The policy network is updated every 100 steps, and when applicable, the target network is also updated every 100 steps. ReLU is used as the activation function for all networks. For exploration, an $\epsilon$-greedy strategy is employed with $\epsilon$ decaying linearly from 1.0 to 0.1. The architectures are as follows: **DQN** uses an MLP with a 64-unit embedding layer and two 64-unit hidden layers; the target network shares this structure. **PPO** uses a shared MLP for policy and value networks with the same layer sizes; Generalized Advantage Estimation (GAE) is applied with $\lambda = 0.95$, the clip ratio is 0.05, and an entropy coefficient of 0.01 encourages exploration. **GCN+DQN** uses a graph convolutional layer for feature extraction, followed by an MLP with two 64-unit hidden layers, mirrored by the target network.

## 5.2   MAIN RESULTS

The experiments were conducted using three algorithms: DQN, GCN+DQN, and PPO, each with or without our DFR framework. This yields six algorithm settings, including DQN+DFR *v.s.* DQN+AD, GCN+DQN+DFR *v.s.* GCN+DQN+AD, and PPO+DFR *v.s.* PPO+AD, where "AD" stands for All Dynamics. Based on these parameter settings (see Section 5.1), we trained the DPP model on three different graphs (see Figure 4) and conducted statistical analysis on key metrics, including Mean GAP, SR, and CR. Since smaller values of GAP and CR indicate better performance, whereas larger SR is preferable, we plot $1 - \text{GAP}$ and $1 - \text{CR}$ to maintain a consistent interpretation.

As shown in Figure 5, each model's performance is visualized as a triangle formed by $1 - \text{GAP}$, SR, and $1 - \text{CR}$, with it area serving as a summary of overall performance. Across all algorithm settings,

---

[4]The source code of all experiments is publicly available at `https://anonymous.4open.science/r/UrbanDynamicPathPlanning-A59E`.

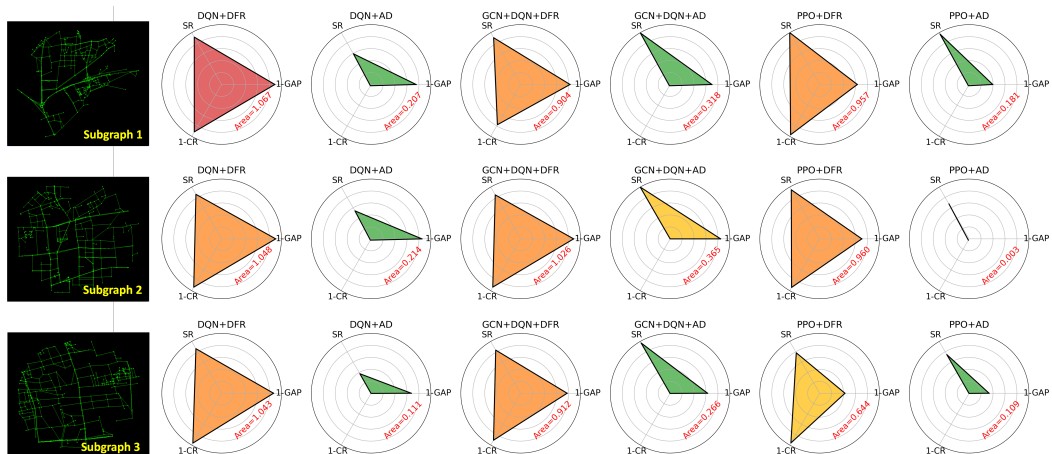

Figure 5: Performance comparison of different algorithms across three regions. "DFR" stands for our framework, while "AD" refers to the use of All Dynamics features from the entire graph.

DFR-enhanced models exhibit larger triangle areas compared to their AD counterparts, indicating superior overall performance. A closer examination reveals that GCN-based models, while achieving high SR, can still exhibit high GAP (i.e., low $1 - \text{GAP}$). This occurs because, although GCN introduces structural representation capability, the combination of a relatively small network and high feature dimensionality limits the model's ability to fully exploit dynamic information. Consequently, the model completes planning tasks but remains largely insensitive to dynamic variations, resulting in high GAP. Incorporating DFR mitigates this limitation and improve the sensitivity of GCN-based models to dynamic changes.

Moreover, DFR substantially reduces feature dimensionality, which corresponds to an increase in $1 - \text{CR}$. This indicates that the model's input dimensionality decreases and the computational overhead for collecting dynamics features is reduced. After incorporating DFR, the average path planning time is $8.18 \pm 1.74$ ms for DQN/PPO and $27.26 \pm 6.8$ ms for GCN+DQN. Correspondingly, DFR reduce the average time of planning paths by 85.59%, 46.08%, and 79.32%, compared to DQN+AD, GCN+DQN+AD, and PPO+AD, respectively. Overall, our DFR framework achieves the fastest planning among RL baselines while maintaining high performance, highlighting that structural representation learning and dynamics feature reduction are complementary.

## 5.3 ABLATION STUDY

We conduct the ablation experiments of $k$ and $n$ on Subgraph 1 (a region of Nanjing; see the left side of Figure 4): 1) $k = \{0.2, 0.4, 0.6, 0.8, 1.0, -1.0\}$, which specifies the proportion of top-100 shortest paths used to construct the subgraph. For instance, $k = 0.6$ selects the top 60 paths, while $k = -1.0$ disables policy attention. 2) $n = \{1, 2, 3, 4, -1\}$ determines the order of neighboring nodes incorporated when constructing local features. For example, $n = 2$ includes 2-hop neighbors, and $n = -1$ corresponds to no hop selection. We use $(-1.0, -1)$ to represent $(k = -1.0, n = -1)$.

The top of Figure 6 shows heatmaps for Mean GAP, SR, and CR across all $(k, n)$ configurations, and several trends can be observed from the experimental results. First, by comparing the baseline (DQN+AD or $k = -1.0, n = -1$) with DQN+DFR ($k > 0.0, n > 0$), the role of DFR in dynamics feature compression can be clearly observed. The baseline exhibits a SR of 0.884 and a Mean GAP of 0.170. With the introduction of DFR and appropriate tuning of $k$ and $n$, the model achieves higher SR and lower Mean GAP and CR. For instance, when $n = 4$ and $k$ is set from 0.4 to 1.0, the Mean GAP values are 0.095, 0.113, 0.108, and 0.108, respectively, while CR remains below 5.7% in all cases. Overall, policy attention selects globally relevant dynamics, whereas $n$-hop neighborhoods captures local dependencies. Together, they enable the DFR framework to improve the model performance while minimizing input dimensionality.

Next, we aim to observe a clear relationship between the degree of neighborhood reduction, the strength of policy attention, and the overall model performance. When $k$ is fixed, for example

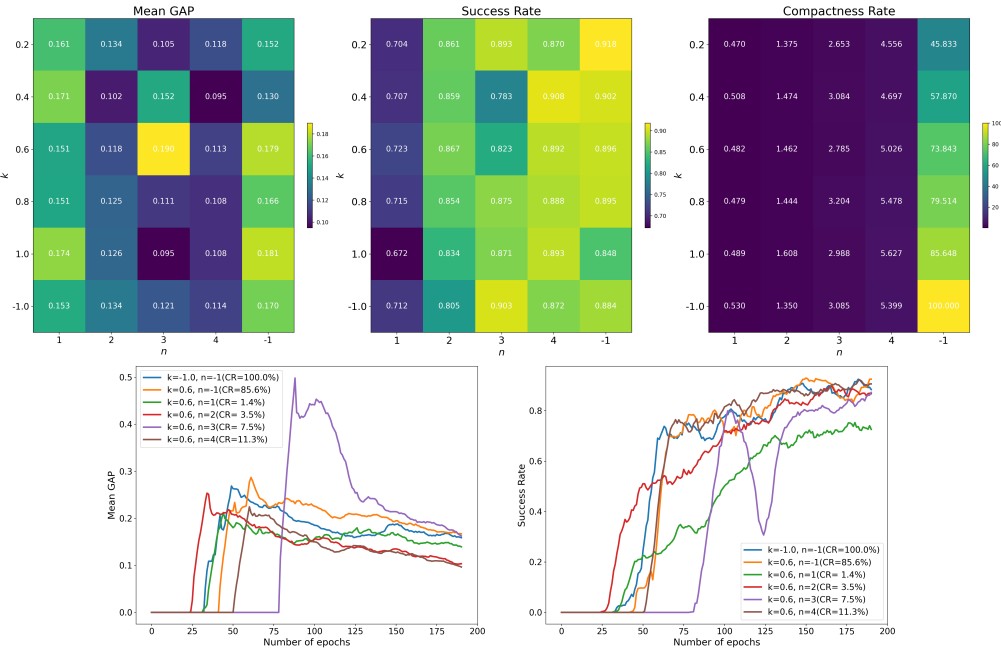

Figure 6: *Top:* Performance headmaps of the models under various combinations of $k$ and $n$ ($k = -1.0, n = -1$ means no policy attention mechanism and no $n$-hop neighborhood selection). *Bottom:* Training curves under $k = 0.6$ with varying $n$.

$k = 0.6$, the model performance is relatively poor when $n$ is small. Specifically, for $n = 1$, the Mean GAP and SR are 0.151 and 0.723, respectively. As $n$ increases, the performance improves noticeably; for instance, when $n = 2$, the Mean GAP decreases to 0.118 and the SR increases to 0.867. However, as shown in the bottom of Figure 6, further increasing $n$ yields limited performance gains, and the curves exhibit a trend of aggregation. For example, when $n = 4$, the Mean GAP and SR are 0.113 and 0.892. On the other hand, when $n$ is fixed, increasing $k$ generally improves performance, but if $k$ exceeds a certain range, performance may decline. For example, when $n = 4$ and $k$ increases from 0.4 to 0.6, the Mean GAP rises slowly from 0.095 to 0.113, while the SR decreases from 0.908 to 0.892. In summary, compared with $n$, $k$ has a more complex and less predictable impact on model performance, which poses greater challenges for parameter tuning. Therefore, based on these systematic conclusions from small-scale experiments, it is recommend that in large-scale graph deployment, configurations with moderate $k$ and smaller $n$ should be preferred. $n$ increases until the aggregation boundary of $n$ is found, and $k$ can be further explored.

## 6 CONCLUSIONS

In this paper, we present a Dynamics Feature Representation (DFR) framework for RL-based dynamic path planning (DPP), which refines global traffic dynamics into compact, node-related local features. By incorporating policy attention mechanism and $n$-hop neighborhood method in DFR, our approach enables the RL agent to effectively capture task-relevant and time-varying dynamics while reducing the dimensionality of state representation. Extensive experiments on realistic urban road networks demonstrate that our DFR framework significantly affects the performance of RL-based DPP model, highlighting the importance of appropriately designing dynamics features. Our study provides insights into how hierarchical feature refinement can improve the adaptability and efficiency of RL-based path planning under dynamic environments. However, the two parameters of $k$ and $n$ in DFR are manually selected in this study, which may limit its practical applicability in real-world urban road networks. In the future, a self-adaptive approach could automatically adjust $k$ and $n$ to allow the DFR framework to dynamically focus on the large-scale road network without manual intervention, thereby enhancing the scalability and real-world deployment potential.

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

## A  USE OF LARGE LANGUAGE MODELS

Large Language Models (LLMs) were used to assist with language polishing, improving clarity and readability of the text. All scientific content, experiments, and results were designed and conducted by the authors.

