# OpenReview forum: "Learning Dynamics Feature Representation via Policy Attention for Dynamic Path Planning in Urban Road Networks"
_ICLR.cc/2026/Conference — ICLR 2026 Poster_

### Official Review · Reviewer_5jVk · 2025-10-30

**Soundness:** 3
**Presentation:** 3
**Contribution:** 2
**Rating:** 6
**Confidence:** 3

**Summary:**

This paper tackles the problem of Dynamic Path Planning (DPP) in urban road networks under time-varying traffic conditions. The authors focus on the challenge of designing effective state representations in reinforcement learning (RL)-based DPP, balancing completeness and efficiency.

They propose a Dynamic Feature Representation (DFR) framework that hierarchically refines global traffic dynamics into compact, decision-relevant features. The approach consists of two stages:

1. Policy Attention: Pre-trained to extract a task-relevant subgraph based on top-k shortest paths;
2. n-hop Neighborhood: Further reduces the representation to local node-centric features.

Experiments on an OpenStreetMap-based Nanjing city network show that DFR improves the convergence speed and path optimality of DQN-based models while significantly reducing feature dimensionality.

**Strengths:**

The paper presents a clear motivation and focuses on an important and practically relevant challenge in dynamic decision-making under uncertainty. The proposed hierarchical state refinement framework is conceptually novel and effectively bridges the gap between global completeness and local efficiency in RL-based path planning. The integration of Policy Attention and n-hop Neighborhood represents an  interpretable way to distill global traffic dynamics into task-relevant local states. Moreover, the authors provide a solid theoretical grounding by connecting their approach to Predictive State Representation (PSR) theory, which strengthens the methodological rigor. The experimental design is systematic, showing how different values of (k, n) influence performance, and the released code enhances reproducibility. Quantitative results show that DFR achieves faster convergence and comparable or improved routing performance with significantly lower feature dimensionality. Overall, the work offers a meaningful step toward scalable RL in real-world traffic environments.

**Weaknesses:**

Despite its conceptual strengths, the paper’s experimental scope is limited to a single city (Nanjing) and relatively small subgraphs, leaving the scalability and generalization of the approach unclear. The baseline comparison is also narrow, relying solely on DQN, without evaluating against more advanced or specialized methods such as GNN-RL, actor–critic algorithms, or model-based RL frameworks. The paper would benefit from more detailed explanations of the policy attention pre-training process, including the dataset, learning stability, and computational overhead. In addition, the hyperparameters k and n are fixed and manually chosen; introducing or discussing a self-adaptive mechanism for these parameters would make the approach more robust. Finally, while the theoretical connection to PSR is strong, there is limited analysis of how the temporal dependencies of traffic dynamics are preserved beyond the spatial context.

**Questions:**

What is the computational overhead when applying DFR to real-world traffic datasets with live updates?

---

> ### Author Response · Authors · 2025-11-21
>
> We addressed the reviewers’ concerns by implementing several major improvements in the manuscript. Below are responses based on specific questions (The line $x$ refers to the $x$-th line in the revised manuscript.):
>
> **Question:** Despite its conceptual strengths, the paper’s experimental scope is limited to a single city (Nanjing) and relatively small subgraphs, leaving the scalability and generalization of the approach unclear.
>
> - To address this concern, we expanded the experimental scope by adding two additional real-world regions: Chaoyang District in Beijing and Pudong District in Shanghai. We conducted the same set of experiments on these new road networks (see line 420-466). It is worth noting that the networks of the new cities are still small-scale. This design choice is intentional: the focus of this study is on developing the DFR framework and establishing its theoretical foundation, specifically the mapping (see line 249) and proposing an instances of $\Psi$ and $\Phi$, corresponding to policy attention and $n$-hop neighbors, respectively. While we acknowledge that these instances are not perfect and require manual tuning of $k$ and $n$, the primary goal of this study is to investigate how different dynamics subsets affect the performance of the DPP model. For this purpose, small-scale urban networks are more suitable. Nevertheless, in future work, we plan to extend the DFR framework to large-scale city networks and develop new instances of $\Psi$ and $\Phi$ that avoids manual parameter tuning as much as possible, enabling practical deployment of DFR framework in real-world DPP scenarios.
>
> **Question:** The baseline comparison is also narrow, relying solely on DQN, without evaluating against more advanced or specialized methods such as GNN-RL, actor–critic algorithms, or model-based RL frameworks.
>
> - We have extended the baseline comparisons to include PPO and GCN+DQN, in addition to the original DQN baseline. For each algorithm, we evaluate the impact of incorporating DFR, resulting in comparisons such as PPO vs. PPO+DFR and GCN+DQN vs. GCN+DQN+DFR. Experiments are conducted across three different scenarios, where we train the DPP model using these algorithms and analyze their performance using the standard evaluation metrics. Detailed experimental settings and quantitative results can be found in line 420-466.
>
> **Question:** The paper would benefit from more detailed explanations of the policy attention pre-training process, including the dataset, learning stability, and computational overhead.
>
> - We provide a more detailed description of the policy attention pre-training process, including the design of the reward function (line 323-340), the stability of the learning procedure (line 341-352), and the associated computational overhead (line 362-367).
>
> **Question:** In addition, the hyperparameters k and n are fixed and manually chosen; introducing or discussing a self-adaptive mechanism for these parameters would make the approach more robust.
>
> - We acknowledge this limitation and have added a discussion in the manuscript. Manually tuning k and n is not intended as a final solution, as it is inefficient and can reduce the practical applicability of our DFR framework. A self-adaptive mechanism for these parameters would be a more robust approach and constitutes a promising direction for future work (line 536-539).
>
> **Question:** Finally, while the theoretical connection to PSR is strong, there is limited analysis of how the temporal dependencies of traffic dynamics are preserved beyond the spatial context.
>
> - We have supplemented the theoretical foundation by analyzing how temporal dependencies of traffic dynamics are captured and preserved within our framework (line 298-305).
>
> **Question:** What is the computational overhead when applying DFR to real-world traffic datasets with live updates?
>
> - In our study, the overall cost primarily consists of: 1) collecting updated dynamic information when new traffic data arrive, and 2) computing the DPP model. DFR reduces this cost by leveraging policy attention and $n$-hop neighborhoods, transforming the original requirement of gathering dynamics over the entire network into gathering information only from small subgraphs. Importantly, these subgraphs can be pre-computed offline (see line 361-367). The policy attention mechanism is derived from a static path planning policy based solely on the road network structure, which does not change over time and can thus be reused offline. Similarly, $n$-hop neighborhoods depend only on the static graph topology and can also be generated offline. We further evaluated the practical runtime impact: after incorporating DFR, the average path planning time is $8.18 \pm 1.74$ ms for DQN/PPO and $27.26 \pm 6.8$ ms for GCN+DQN (see line 461-464).

---

> > ### Author Response · Authors · 2025-12-01
> > **Brief summary of our responses to Reviewer 5jVk**
> >
> > We have addressed the reviewers’ concerns by implementing several major improvements in the revised manuscript. First, we broadened the set of baseline algorithms by adding PPO and GCN+DQN, which allows evaluating the impact of DFR across different RL frameworks. Second, we added two additional real-world road networks, namely Beijing Chaoyang and Shanghai Pudong, and conducted experiments with all baseline and DFR-enhanced algorithms. Third, we provided more comprehensive details on experimental configurations, training protocols, and result analyses, offering deeper insight into the DFR framework’s contributions. In addition, in response to the reviewer’s suggestions regarding the description of PSR in the Methodology section, we have provided further clarification and supplementary explanation.

---

### Official Review · Reviewer_ZKPV · 2025-10-31

**Soundness:** 3
**Presentation:** 3
**Contribution:** 3
**Rating:** 6
**Confidence:** 3

**Summary:**

This paper focuses on the trade-off between completeness and efficiency in reinforcement learning state representations for urban dynamic path planning (DPP). It proposes a hierarchical dynamic feature representation framework, DFR, which combines policy attention with an n-hop neighbourhood approach to effectively compress the state space while enhancing policy performance.

**Strengths:**

1. High innovativeness: The proposed DFR framework integrates policy attention with n-hop neighbourhoods, pioneering the combination of ‘task-relevant graph sparsification’ and ‘local dynamic feature extraction’ for DPP problems. The introduction of Predictive State Representation (PSR) theory enhances interpretability and theoretical grounding.

2. Methodological Rationality: The policy attention mechanism extracts task-relevant subgraphs via a pre-trained shortest path policy, effectively reducing redundant information. The n-hop neighbourhood approach further localises states, balancing computational efficiency with information integrity.

**Weaknesses:**

1. Methodology Explanation and Experimental Validation

2. Generalization Capability of the Model

3. Insufficient Completeness of Experimental Results

**Questions:**

1. Methodology Explanation and Experimental Validation:
The experimental evaluation appears to test only fixed combinations of k and n values. It remains unclear whether the method dynamically adjusts these parameters across cities of varying scales or under different traffic densities. The paper would benefit from a more in-depth discussion on the tuning of hyperparameters k and n—specifically, how to select these parameters to achieve optimal performance under diverse network structures and traffic conditions. Furthermore, is there a method to automate the selection of these parameters to minimize manual configuration efforts? Addressing this would significantly enhance the practicality and scalability of the proposed approach.

2. Generalization Capability of the Model:
The current experiments are conducted solely on a road network subgraph from Nanjing, which consists of a relatively small scale (378 nodes and 864 edges). This raises concerns about whether such a subgraph is representative of larger and more complex urban traffic systems. It is essential to evaluate the model's performance on road networks from other cities, particularly on larger-scale networks (e.g., thousands of nodes and tens of thousands of edges). Further experiments across cities with varying traffic conditions and network topologies are recommended to verify the model's universality and robustness.

3. Insufficient Completeness of Experimental Results:
The paper only compares the proposed method against baseline models such as "No Policy Attention" and "No Neighborhood Expansion," which are essentially ablative studies. There is a lack of comparison with other established path planning methods, especially traditional dynamic programming approaches in dynamic environments.

---

> ### Author Response · Authors · 2025-11-21
>
> In response to the reviewers’ feedback, we have substantially enhanced the manuscript. Below are responses based on specific questions (The line $x$ refers to the $x$-th line in the revised manuscript.):
>
> **Question:** Methodology Explanation and Experimental Validation: The experimental evaluation appears to test only fixed combinations of k and n values ... Furthermore, is there a method to automate the selection of these parameters to minimize manual configuration efforts? Addressing this would significantly enhance the practicality and scalability of the proposed approach.
>
> - We thank the reviewer for this insightful comment. When applying DFR to different cities, $k$ and $n$ need to be adjusted to accommodate varying road network structures and traffic conditions. Beyond the original Nanjing network, we have added two additional regions—Chaoyang District in Beijing and Pudong District in Shanghai (see line 373-376 and Figure 4 at line 330)—and varied the traffic factor $\beta$ (see Equation 9 at line 386 in the revised manuscript) to simulate different traffic densities. In the revised manuscript, we have merged the original main results with the ablation study. Using the new baseline algorithms across three scenarios, we conduct comparative experiments, and the analysis results are now presented as the updated main results. In the new ablation study section (line 469-524), we perform a detailed investigation of $k$ and $n$ from two perspectives: the effectiveness of DFR itself and its influence on model performance. Based on this analysis, we provide empirical guidelines for adjusting $k$ and $n$ under different conditions. In addition, we acknowledge that manually tuning these parameters is not ideal. We agree that automating the selection of $k$ and $n$ would enhance scalability and practicality, and developing such adaptive mechanisms is a promising direction for future work (see line 536-539).
>
> **Question:** Generalization Capability of the Model: The current experiments are conducted solely on a road network subgraph from Nanjing ... Further experiments across cities with varying traffic conditions and network topologies are recommended to verify the model's universality and robustness.
>
> - We thank the reviewer for raising this important point. To evaluate the model’s generalization capability, we have expanded our experiments by including two additional real-world regions: Chaoyang District in Beijing and Pudong District in Shanghai. We conducted the same set of experiments across these three cities, covering different traffic conditions and network topologies. The results demonstrate consistent performance trends, indicating that our approach generalizes well across diverse urban environments (see line 420-466).
>
> - As for the large-scale scenario, we have provided corresponding explanations and discussions (see line 536-539): The primary focus of this work is on developing the DFR framework and establishing its theoretical foundation, specifically the mapping (see line 249). The policy attention mechanism and $n$-hop neighborhood method presented in this paper serve only as feasible instances of $\Psi$ and $\Phi$; they are not intended to be optimal. The main goal is to systematically investigate how different dynamic subsets affect the performance of the DPP model. In this context, small-scale networks are more suitable, as they allow efficient exploration and controlled analysis of parameter effects. In future work, the insights and empirical observations obtained from small-scale experiments will be extended to large-scale networks. They will also guide the design of new instances of $\Psi$ and $\Phi$, such as approaches that automatically select the most informative dynamic subsets based on local and global traffic dynamics, enabling practical deployment of DFR in complex urban traffic systems.
>
> **Question:** Insufficient Completeness of Experimental Results: The paper only compares the proposed method against baseline models such as "No Policy Attention" and "No Neighborhood Expansion," which are essentially ablative studies. There is a lack of comparison with other established path planning methods, especially traditional dynamic programming approaches in dynamic environments.
>
> - To provide a more comprehensive comparison, we have extended our experiments beyond baselines. In addition to the original DQN baseline, we include PPO and GCN+DQN as representative RL-based DPP algorithms. For each algorithm, we evaluate the effect of incorporating DFR, resulting in comparisons such as DQN vs. DQN+DFR, PPO vs. PPO+DFR and GCN+DQN vs. GCN+DQN+DFR. Experiments are conducted across three urban scenarios (see line 420-466).

---

> > ### Author Response · Authors · 2025-12-01
> > **Brief summary of our responses to Reviewer ZKPV**
> >
> > In response to the reviewers’ feedback, we have improved our manuscript. First, we have incorporated the reviewer’s suggestions by adding further clarification of the theoretical foundations in the Methodology section. Second, we expanded the experimental evaluation to include two additional RL algorithms, PPO and GCN+DQN, enabling comprehensive comparisons such as PPO vs. PPO+DFR and GCN+DQN vs. GCN+DQN+DFR. Finally, to demonstrate the generalization and scalability of our framework, we incorporated two additional urban road networks—Chaoyang District in Beijing and Pudong District in Shanghai—and conducted experiments across all algorithms, providing detailed analyses and visualizations in the revised Main Results section.

---

### Official Review · Reviewer_yz7p · 2025-10-31

**Soundness:** 2
**Presentation:** 2
**Contribution:** 1
**Rating:** 2
**Confidence:** 5

**Summary:**

This paper introduces a Dynamic Feature Representation (DFR) framework for Reinforcement Learning (RL)-based Dynamic Path Planning (DPP). The DFR framework refines complex global traffic dynamics into compact, decision-relevant local features by incorporating a policy attention mechanism and n-hop neighborhood methods. This significantly improves the adaptability and efficiency of RL agents in dynamic traffic environments. Extensive experiments demonstrate that DFR improves the performance of RL-based path planning models and reduces computational overhead, offering an efficient solution.

**Strengths:**

1) The DFR framework combines reinforcement learning and dynamic traffic information representation, proposing a new feature representation method.

2) The figures and equations are intuitive and easy to understand, effectively conveying the main content of the research.

**Weaknesses:**

1)	I think the areas that need improvement in the paper are mainly in the experimental section. Firstly, the experiments only compare the performance of the DFR framework with different k and n values, without comparing it to other advanced dynamic path planning algorithms (such as other RL algorithms), which reduces the convincing power of the method's effectiveness.

2)	Secondly, there are missing details in the experimental design and analysis, such as the training details of the model and the analysis of the experimental results in Figure 5.

**Questions:**

1.	In the "Model Training" section of 5.1, could the loss function be introduced? This would help readers understand the model’s learning process.

2.	Regarding the experiments, since the authors did not compare their method with other advanced algorithms and only demonstrated the effectiveness of their algorithm through the comparison of (-1.0, -1) in 5.2.  It should be noted that (-1.0, -1) means no policy attention mechanism and no n-hop neighborhood selection. Based on my understanding, this essentially serves as an extreme baseline setup, but more baseline descriptions are needed to justify the reasonableness of this comparison.

3.	In line 421, "When n = 4, the model achieves a lower mean GAP than the baseline while maintaining a comparable SR," which baseline does this refer to? Is it (-1.0, -1) or (0.6, -1)? The significance of this result can be further analyzed rather than just describing the observed outcome.

4.	In model training, the experiment is set to 600 episodes, but Figure 5 only shows the results for the first 175 episodes. Why weren’t the later episodes’ results displayed?

5.	The title and axis labels in Figure 6 are incorrect.

6.	In the ablation study section, considering that there are some label errors in Figure 6, is this the result after 600 full episodes of training? I believe Figure 6 actually conveys the same meaning as Figure 5, and the analysis in this section seems to be redundant. The overall experimental results appear more like an ablation study, without fully highlighting the advantages of the algorithm.

---

> ### Author Response · Authors · 2025-11-21
>
> In response to the weaknesses and questions identified by the reviewer, we made several substantial improvements to the manuscript, especially in the experiment section. Below are responses based on specific questions (The line $x$ refers to the $x$-th line in the revised manuscript.):
>
> **Question:** In the "Model Training" section of 5.1, could the loss function be introduced? This would help readers understand the model’s learning process.
>
> - We have added the definition of $Q$ function (see line 188 and Equation 3) and the Temporal-Difference (TD) error $\mathcal{L}$ (see Equation 10 at line 402) as the loss function to clarify the learning process of the DPP model in the "Model Training" section. Specifically, the model updates its $Q$-function by minimizing $\mathcal{L}$.
>
> **Question:** Regarding the experiments, since the authors did not compare their method with other advanced algorithms and only demonstrated the effectiveness of their algorithm through the comparison of (-1.0, -1) in 5.2. It should be noted that (-1.0, -1) means no policy attention mechanism and no n-hop neighborhood selection. Based on my understanding, this essentially serves as an extreme baseline setup, but more baseline descriptions are needed to justify the reasonableness of this comparison.
>
> - The reviewer’s comment regarding the extreme baseline $(−1.0,−1)$ is highly professional and reasonable. However, the primary goal of our experiments is to systematically obtain the relationship between parameters $k,n$ and DPP model performance. The experiments are conducted on relatively small maps, which allows for efficient parameter tuning. Insights gained from these small-scale experiments are then used to guide the configuration and deployment of the DFR framework on larger maps. To further mitigate the potential limitations of this extreme baseline setting, we have supplemented comparative experiments to evaluate the effect of DFR across different RL paradigms and representation strategies, including DQN vs DQN+DFR, PPO vs PPO+DFR, and GCN+DQN vs GCN+DQN+DFR (see line 420-466). The results clearly demonstrate the effectiveness of the DFR framework across multiple baseline algorithms.
>
> **Question:** In line 421, "When n = 4, the model achieves a lower mean GAP than the baseline while maintaining a comparable SR," which baseline does this refer to? Is it (-1.0, -1) or (0.6, -1)? The significance of this result can be further analyzed rather than just describing the observed outcome.
>
> - Although both $(−1.0,−1)$ and $(0.6,−1)$ appear consistent with the statement in the figure, the intended baseline here is $(0.6,−1)$. The analysis in this part focuses on fixing $k=0.6$ and varying $n$ to examine the effect of $n$-hop neighbors method. We acknowledge that the original explanation was not sufficiently clear. In the revised version, this analysis has been integrated into the ablation study section, where we provide a more thorough discussion from two perspectives: the effectiveness of DFR itself and the relationship between DFR configurations and performance improvements (see line 469-524 for details).
>
> **Question:** In model training, the experiment is set to 600 episodes, but Figure 5 only shows the results for the first 175 episodes. Why weren’t the later episodes’ results displayed?
>
> - The x-axis label in Figure 5 in the original paper is incorrect. It is mistakenly marked as *the number of episodes*, but it actually represents *the number of epochs*. The model is trained for 75,600 steps, and evaluation is performed periodically during training, producing a total of 200 epochs of test results. We have corrected the x-axis label in the revised figure (see Figure 6 at line 500) and added a clarification note in line 408 of the revised manuscript.
>
> **Question:** The title and axis labels in Figure 6 are incorrect.
>
> - We have corrected the title and axis labels in Figure 6 (see line 490) to accurately reflect the corresponding metrics and settings.
>
> **Question:** In the ablation study section, considering that there are some label errors in Figure 6, is this the result after 600 full episodes of training? I believe Figure 6 actually conveys the same meaning as Figure 5, and the analysis in this section seems to be redundant. The overall experimental results appear more like an ablation study, without fully highlighting the advantages of the algorithm.
>
> - We acknowledge that Figures 5 and 6 in the original manuscript essentially present ablation studies. To clarify and streamline the presentation, we have made the following modifications: 1) Figures 5 and 6 in the original manuscript have been merged into Figure 6 (see line 486-506) in the revised manuscript, and their content is now fully incorporated into the ablation study section (see line 469-524); 2) we have added a comparison against baseline algorithms, which is presented as the main results to more clearly highlight the advantages of our proposed method (see line 420-465).

---

> > ### Author Response · Authors · 2025-12-01
> > **Brief summary of our responses to Reviewer yz7p**
> >
> > In response to the weaknesses and questions identified by the reviewer, we made several substantial improvements to the manuscript, especially in the experiment section. First, we extended the experimental comparisons beyond the original DQN baseline by including two additional representative RL algorithms: PPO and GCN+DQN, resulting in experiments such as PPO vs. PPO+DFR and GCN+DQN vs. GCN+DQN+DFR. Second, to demonstrate the generalization and scalability of our framework, we incorporated two additional real-world urban road networks, namely the Chaoyang District of Beijing and the Pudong District of Shanghai, and conducted experiments with all the aforementioned algorithms, providing detailed visualizations and analyses of the results in the revised Main Results section. Third, the original Main Results section was merged into the Ablation Study section, creating a more coherent and integrated presentation of the experimental findings. Finally, we supplemented additional details regarding the experimental setup, training parameters, and analysis of results, ensuring a clearer understanding of the model’s learning behavior and the impact of the DFR framework.

---

### Meta-Review · Area_Chair_ndkh · 2026-01-08

**Summary:**

This paper proposes a Dynamic Feature Representation (DFR) framework for RL-based dynamic path planning in urban road networks, aiming to balance state completeness and computational efficiency. Reviewers generally appreciated the clear motivation, interpretable design combining policy attention and n-hop neighborhoods, and the theoretical grounding via predictive state representation. Positive feedback highlighted the potential of DFR to accelerate convergence and improve planning performance while reducing feature dimensionality.

The main concerns affecting the decision stemmed from the initially limited experimental scope and baseline coverage. One reviewer expressed strong skepticism due to the lack of comparisons with other RL algorithms and unclear experimental presentation, while others raised concerns about generalization beyond a single city, scalability, manual hyperparameter tuning (k, n), and insufficient experimental details. Overall, the work was viewed as conceptually sound but initially borderline due to experimental incompleteness rather than flaws in the core idea or methodology.

**Reviewer Concerns:**

Most major reviewer concerns were substantially addressed in the rebuttal and revised manuscript. The authors expanded experimental evaluations to include multiple RL baselines (DQN, PPO, GCN+DQN), added two additional real-world urban road networks (Beijing and Shanghai), corrected figure and labeling issues, and significantly improved experimental clarity and analysis. Requests for stronger baselines, clearer training details, and broader generalization evidence were directly addressed.

Some concerns remain partially open. In particular, the framework still relies on manual tuning of hyperparameters (k, n), and experiments are conducted on relatively small-scale road networks. While the authors provide empirical guidelines and clearly acknowledge automation and large-scale deployment as future work, these limitations remain inherent to the current submission. Nevertheless, the remaining issues are appropriately scoped and do not undermine the technical validity of the proposed framework.

**Reviewer Scores:**

Reviewer yz7p (initial: 2): Likely increases to 4–5, given the addition of multiple baselines, new cities, corrected experiments, and improved clarity, though may still hold reservations.

Reviewer ZKPV (initial: 6): Likely remains 6 or increases to 7, as concerns on generalization and baselines were addressed.

Reviewer 5jVk (initial: 6): Likely remains 6 or increases to 7, with expanded experiments and clearer methodological explanations.

One reviewer withdrew, reducing its impact on the final assessment.

Overall post-discussion sentiment trends positive.

---

### Decision · Program_Chairs · 2026-01-26

Accept (Poster)